# Private Health Insurance and Financial Risk Taking in Spain—The Moderating Effect of Subjective Risk Tolerance

**DOI:** 10.3390/ijerph192316248

**Published:** 2022-12-05

**Authors:** Inmaculada Aguiar-Díaz, María Victoria Ruiz-Mallorqui

**Affiliations:** Department of Financial Economics and Accounting, Faculty of Economy, Business and Tourism, University of Las Palmas de Gran Canaria (Spain), 35001 Las Palmas, Spain

**Keywords:** private health insurance, financial risk taking, financial risk tolerance, household finance, Spain

## Abstract

This study focuses on the effect of private health insurance on financial risk taking in Spanish households. According to the arguments related to the background risk, we propose two hypotheses: the first predicts a positive relationship between health insurance and risk taking and the second asserts that attitude to risk moderates this relationship. Spain is a good laboratory because it has a National Health System (NHS) that offers healthcare coverage to the entire population, which could eliminate the effect of health insurance on risk taking. Based on a sample of 6110 households obtained from the Household Finance Survey (EFF), our results confirm both hypotheses. Specifically, we show that private health insurance significantly increases a household’s portfolio risk, especially in households with greater risk aversion. The results are concordant with the scarce amount of previous empirical evidence obtained in other contexts and are robust for different subsamples and estimation methods.

## 1. Introduction

The unpredictable nature of health problems means that the costs derived from health services represent two related, although different, sources of background risk: health risk and medical expenditures risk [1,2]. The first depends only on health status, while the second is a function not only of health risk, but also of health insurance coverage [2]. The public system and/or the private sector can finance this coverage. The literature distinguishes between three motivations for acquiring private health insurance (PHI): (i) to complement public health coverage, (ii) to obtain the provision of health services when public coverage is not available, or (iii) a combination of both. In countries with a financed National Health System (NHS) that grant a basic package of medical expenditures, like Spain, PHI provides additional, specific services or is contracted to reduce waiting lists [3]. Waiting to receive treatment has a cost in terms of health. The longer the wait, the higher the possibility of damage to health and the lower the probability of full recovery after the procedure [4].

From the perspective of household finances, PHI has two related effects. First, it reduces the risk derived from health expenses, which contributes toward improving the quality of life of individuals and their families. Second, this coverage of medical expenses increases available resources, which could be used to make financial investments, positively affecting the household’s financial portfolio. Therefore, PHI is not only a protection mechanism for health, but also for family finance [5]. In the same vein, Li et al. [6] assert that medical expenditure risk is widely believed to reduce the willingness of households to bear other risks and, in turn, could alter their behaviour. In this sense, the security offered by PHI can lead to risky financial decisions that potentially offer better returns, and that such protection facilitates the availability of resources for riskier financial investments. Likewise, the adoption of financial decisions should be in accordance with the risk tolerance of individuals. Thus, investment in high-risk financial products generates uncertainty regarding the results, which may affect the level of stress depending on the investor’s degree of subjective risk tolerance. Thus, it is interesting to analyse the moderating effect of the risk tolerance in the relationship between PHI and financial risk taking. In this context, the objective of this work is to determine the effect of PHI on financial risk taking in Spanish households, considering the possible moderating effect of risk tolerance.

Our work presents several contributions. The first focuses on analysing the effect of PHI on household financial risk taking in Spain, a country with a NHS, in which—unlike the USA for example—the entire population has public health coverage. So far, the only studies that have analysed the relationship between private health insurance and portfolio risk refer to the USA [2,7] and China [6]. In this sense, Spain is a good laboratory since it has public health coverage that could eliminate the effect of PHI on risk taking. Second, our sample includes a wide range of ages; not only older people. That sample allows us to consider the effect of age on families’ financial investment decisions, which has been an important issue analysed in the life cycle literature. The two studies mentioned above [2,7] use an American survey of individuals aged 50 and over. Third, we consider the moderating effect of subjective risk tolerance, which has only previously been analysed in the context of China. Fourth, in our empirical analysis, we rely on PHI and self-reported measures of attitude to risk in regards to portfolio allocations. To the best of our knowledge, in the European context, data on these variables are separately available in several household surveys, but they are treated together only in the Survey on Health, Ageing and Retirement in Europe (SHARE) and the Household Financial Survey (hereinafter, EFF for its Spanish acronym). However, SHARE contains a sample of households whose head is aged 50 or older, while the EFF includes a sample of people from 18 to 85 years. For this reason, EFF is an ideal source of data for our purpose.

The remainder of this paper is organised as follows. In the following section, we provide a summary of the Spanish healthcare system. The third section contains the related literature and presents the hypotheses. In the fourth, we outline the methodology followed in the empirical study and in the fifth, we present the results. In the sixth, we discuss our results based on the little previous empirical evidence. Lastly, we summarise the conclusions of the study.

## 2. Private Health Insurance in Spain

In European countries, even in those with an NHS, medical expenditure risk has increased due to a reduction in public spending on healthcare as a result of the imposition of the Maastricht Treaty to limit the public deficit, as well as the 2008 economic crisis. Consequently, many Europeans have been faced with cost-cutting measures that have led to reduced provision of public health services, increased cost-sharing and increased provision of private sector insurance [1].

In this context, Spain belongs to the group of European countries with a NHS, where 99% of the population has access to a public healthcare system, financed mainly through taxes [8], and compulsory contributions from working citizens, regardless of whether they have taken out private health insurance. In this country, PHI therefore has a double role: (i) providing additional coverage for services already covered by NHS and (ii) providing coverage for services not covered by the NHS, such as dental care for adults [9]. Therefore, according to Pinilla and López-Valcarcel [9], in Spain the private health insurance is supplementary or alternative to the coverage offered by public health. In addition, although the existence of universal coverage allows citizens to use public services free of charge, it is not possible to request specialised consultation directly, this having to be done through the family doctor or general practitioner (GP). This GP then refers the consultation to the system and enters it into the queue, where it is placed depending on the severity. In consequence, it is not possible to choose the specialist who is assigned by the healthcare system. Therefore, PHI can allow individuals access to an elected medical specialist.

From an organisational point of view, in Spain, health-related competencies have been transferred to the autonomous communities, which are in charge of managing financial and human resources, as well as agreements with private entities to reduce waiting lists and times. This sometimes entails long waiting lists to receive certain treatments, medical consultations or surgical interventions. Thus, PHI is usually bought to avoid waiting times in the public NHS. In this sense, Jofre-Bonet [4] finds a positive relationship between a lower quality of public healthcare (longer waiting times) and a higher probability of buying private insurance in Spain. According to the 2018 Healthcare Barometer [10], 78% of individuals in Spain with double coverage declared that their main reason for buying private health insurance was the waiting time [9]. For this reason, Spaniards resort to contracting PHI in order to obtain faster medical assistance, which allows them to improve their quality of life.

Regarding drug expenses, since the Spanish NHS covers a high percentage of the cost of medicines, it should be noted that it is possible to ask the family doctor to prescribe a drug that has been prescribed by a private specialist. Therefore, it is possible to cover such expenses through the public system. Finally, another feature of the Spanish NHS is the possibility for civil servants belonging to the Mutualidad General de Funcionarios Civiles del Estado (MUFACE) to choose between public and private health care, under the agreement between MUFACE and the insurance companies. In this case, civil servants do not have to pay the insurance, as it replaces public health care, covering all medical expenses.

## 3. Literature Review

The theoretical literature demonstrates how portfolio decisions depend on factors such as risk aversion and investment opportunities. The term “risk aversion” and its opposite, “risk tolerant”, are used interchangeably. A fundamental aspect of portfolio theory is the notion that risk and returns are positively related [11]. However, households do not always follow portfolio theory because of—according to Campbell [12]—one, standard finance theory, which describes the wealth-maximising choices of households (normative household finance) and, two, behavioural finance theory, which describes the choices actually made by households (positive household view). In addition, this author indicates that household financial decisions have many special characteristics that could explain the difference between ideal and actual household behaviour. In this sense, one of the aspects that could influence household portfolio allocation is a variety of “background risk” factors, including those related to health and medical expenditure risks [2].

### 3.1. Private Health Insurance and Financial Risk Taking

The existence of risks that cannot be avoided or fully insured against—that is, background risk—make individuals less susceptible to bearing other types of risk [13,14,15]. The presence of background risk leads households to increase precautionary savings, which reduces risky asset holdings [16]. In this sense, Ayyagari and He [17] assert, “economic theory suggests that as background risk decreases, investment in risky financial assets will increase even when the two risks are independent of each other” (p. 1456).

Health represents one of the main sources of background risk, deriving from unexpected medical expenses. Consequently, a medical expenses risk is generated, which could affect the way in which households are willing to take on financial risk in order to boost their investment portfolios [2]. As Ayyagari and He [17] indicate, medical expenditures are a type of un-diversifiable background risk, due to deductibles and other cost-sharing mechanisms, but can be partially mitigated by health insurance coverage.

Logically, medical expenses risk is related to health risk. Thus, some of the literature has analysed the effect of health risk on portfolio choices. Specifically, some authors have found that the higher the health risk (approximated by health status), the greater the proportion of financial wealth invested in safer assets, especially for older people (e.g., [1,18,19]). However, other authors found that poor health has either no direct effect or only a small direct effect on risky portfolio allocation (e.g., [19,20]).

It is possible that the discrepancies observed are due to the failure to consider coverage through PHI. In this sense, the only studies that have considered this aspect are Goldman and Maestas [2], Christellis et al. [7], and Li et al. [6]. Goldman and Maestas [2], in the USA, found that individuals who face lower medical expenditure risk, as measured by their enrolment in a Medicare Health Maintenance Organization (HMO) or an employer supplemental insurance policy, are more likely to hold risky financial assets. Christellis et al. [7] found that Medicare eligibility has a significant impact on stockholding for college-educated households, but do not alter the financial risk-taking behaviour of households whose members have not finished college. Li et al. [6], using a sample obtained from a survey of Chinese households, found that the use of health insurance with better policies is related to a higher probability of acquiring risky assets.

According to the arguments and empirical evidence, the first hypothesis is as follows:

**Hypothesis** **1 (H1).**
*Households with Private Health Insurance Show a Higher Degree of Financial Risk Taking.*


### 3.2. Moderating Effect of Risk Tolerance on the PHI and Risk Taking Relationship

Prospect theory [21] states that individuals have different perceptions of their possible losses and potential gains, which creates biases in behaviour, especially aversion to losses. In other words, they have a greater sensitivity to losses than to profits. Park and Yao [22] define risk aversion as “how much households avoid risk” and, conversely, risk tolerance as “how much households accept risk” (p. 626). Therefore, the term “risk averse” is the opposite of “risk tolerant”, but they are used interchangeably in the finance literature.

More specifically, financial risk tolerance can be defined as a person’s willingness to engage in a financial behaviour in which the outcomes are both unknown and potentially negative [23]. Undoubtedly, attitude toward risk has an impact on an individual’s economic decisions in different contexts [6]—consumption, savings, investments, etc.—and plays a vital role in the individual’s well-being [24]. Specifically, financial risk tolerance plays a key role in shaping decisions on how to allocate financial assets (e.g., [12,23,25]) and, consequently, financial decisions should be in accordance with the risk tolerance of individuals.

The literature has analysed the relationship between risk tolerance and portfolio risk and concluded that it is positive. Among these studies, works based on the Survey Consumer Finance (SCF) in the USA [26,27], along with others in the USA and China [28], predominate. In the European context, Bucciol et al. [29] analysed the link between individuals’ attitudes to financial risk and their investments in risky assets. This author found that individuals who display risk tolerance often decide to buy risky assets, revealing differences between Scandinavian and Mediterranean countries. However, this work used the Survey on Health, Ageing and Retirement in Europe (SHARE), which only includes older individuals (at least 50 years).

To the best of our knowledge, the only study that analyses the moderating effect of risk tolerance in the relationship between health insurance and portfolio risk is Li et al. [6], based on a Chinese household survey. They indicate that attitude to risk is a key aspect of portfolio allocations and insurance policy choices, and that, moreover, “the relationship between health insurance participation and risky asset ownership may also vary by risk preference” (p. 1240). According to their argument, it is important to take into account attitudes to risk when analysing household behaviour in the medical and financial markets. Specifically, they demonstrated that Chinese households who are less risk averse show greater sensitivity to risk substitution and reducing their exposure to medical expenditure risk will result in a significant increase in their willingness to take financial risks. However, for households with high risk aversion, even if they participate in the insurance scheme, they are still unwilling to bear any risk due to their low risk tolerance.

Along this line, it is interesting to explore the possible moderating effect that risk tolerance may have on the relationship between health insurance coverage and the choice of higher or lower risk assets in different contexts, given the idiosyncrasies of each area or country. In this sense, regarding Europe, Bucciol et al. [29] found a wide variation in attitude to risk across both countries and households. Additionally, Georgarakos and Pasini [30] showed, using the same survey of households aged 50 and older across Europe as Bucciol et al. [29], that there are remarkable differences in stock market participation across groups of European countries, considering households with similar net wealth. Those authors also concluded that the variation in attitude to risk helps to explain the large cross-country discrepancies in financial decisions in Europe.

Therefore, although according to hypothesis 1 we expect that maintaining private health insurance will increase financial risk taking, following Li et al. [6], we take into consideration that attitudes to risk may lead to biased estimates. Therefore, it is likely that the degree of risk tolerance affects this relationship and predictable that health insurance protection will have a greater impact on financial risk taking in the case of individuals with greater risk aversion. Consequently, the second hypothesis is enunciated as follows:

**Hypothesis** **2 (H2).**
*Financial Risk Tolerance Moderates the Relationship between Having Private Health Insurance and Financial Risk Taking.*


## 4. Materials and Methods

### 4.1. Sample and Source of Information

The Survey of Household Finances was used as the source of information for this study. The Bank of Spain conducts the EFF on a three-year basis, the most recent corresponding to 2017, although the microdata was available from December of 2020 [31]. The survey breaks down the data into numerous modules, among which are financial investment, health status, health insurance policies, income, labour status, and demographic variables. The population framework for the EFF-2017 sample was the January 2017 Population Register, in which the units are households defined by their postal address. An important characteristic of this sample is the over-representation of wealthy households, which ensures that there is a sufficient number of observations to study the financial behavior of households. The information was collected via personal interviews with the households, conducted between October 2017 and the beginning of June 2018. The EFF corresponding to 2017 contains responses from 6413 households. Some observations were lost because there was no response regarding one of the household’s incomes, which reduced the sample to 6401 observations. Given that the objective of the study focuses on financial risk taking, we have only considered households that have declared holding some type of financial investment, including bank accounts [26]. Our final sample comprises 6110 observations, with full information on our variables of interest.

### 4.2. Variables

Dependent variables. Following the literature, we consider three variables to proxy risk taking: risky asset holdings, portfolio share in risky assets or risky assets ratio, and portfolio share in stock. All variables were obtained from the financial portfolio allocation, considering only the financial assets [2,26,29]. Specifically, we consider bank accounts, savings or term deposits in credit institutions, bonds or fixed income securities (public and private), participation in investment funds or other collective investment entities, (excluding pension funds), stocks in firms (listed and unlisted), and derivative products as financial assets. Risky assets include bonds, investment funds, stocks, and derivatives products. In Table A1 of the Appendix A, we can see the EFF question corresponding to each asset type.

Risky assets holding (RAH) is a dummy variable that takes the value 1 if there are any risky assets in the household portfolio and 0 otherwise [2,6,29]. The Risky assets ratio (RAR) is the share of risky assets in household portfolios, when “risky asset” is computed aggregating the amount invested on bonds, mutual funds, stock and derivatives (e.g., [6,29]). The Portfolio share in stocks (PSS) is the share of financial assets held in stocks (e.g., [20,32,33]). Both variables adopt values from 0 to 100, where more value indicates higher financial risk taking.

Explanatory variables. According to the first hypothesis, PHI is a dummy variable, which adopts the value 1 if the respondent asserts that he has contracted a private health insurance and 0 otherwise. This is based on the responses to question P.5.22 of the EFF. To contrast the second hypothesis, we classify the respondents according their self-assessments of attitude to risk [6,22,27,34,35,36]. Specifically, we create the Subjective risk tolerance (SRT) variable from the answers to question 9.11 of the EFF regarding respondents’ willingness to assume financial risks when they save or make an investment. Based on the answers, we classify the respondents into two groups. The first is integrated for the respondents not willing to take a financial risk (SRT = 0), and the second for the respondents willing to take few, some or many risks (SRT = 1). In addition, we create an interaction variable PHIxSRT, which adopts the value 1 if the respondent is willing to take financial risks and has PHI, and 0 otherwise.

Control variables. Following the literature on health insurance and portfolio allocation [1,2,29,37], health status, gender, age, education level, marital status, presence of children, labour status, income level, and home-ownership are considered as control variables. The description of these variables appears in Table 1.

### 4.3. Estimation Method

The estimation method depends on the dependent variable. Thus, when this was a dummy variable, as with Risky assets holding, we used a Probit method. When the dependent variable was risky assets ratio or portfolio share in stock, we used the Tobit method (e.g., [18,19,44]). We made this decision because in the sample there is a significant share of households with zero risky assets, which is normal according to the studies on household finances [20]. As a robustness analysis, we used an alternative method to control the zeros—the Heckman two-step model. This model provided the results for the two stages. In the first stage, the dependent variable was a dummy that took the value 1 if the household had risky financial assets; the second stage explains the risk asset ratio. In addition, we estimated a simultaneous equation model (3SLS) to jointly estimate the equations for portfolio risk and health insurance [6].

## 5. Results

### 5.1. Descriptive Analysis

The descriptive statistics of variables are shown in Table 2. From this table it can be deduced that 34.81% of the sample have risky assets, while the average percentage of risky assets over the value of the portfolio is 20.54%, and 14.72% of households have investment in stocks. Thirty-three percent of households have PHI and present an average of 2.8 out of 4 in their reported health status. Regarding subjective risk tolerance, only 24.51% affirm that they are willing to assume some risk in exchange for obtaining a return on their financial investments.

The average of each of the three indicators of financial risk taking by PHI and SRT is presented in Table 3. According to the data in this table, households with PHI show a higher degree of risk taking than households without PHI. These differences are statically significant at 1%.

Finally, Table A2 in the Appendix A contains the correlations matrix and the Variance Inflation Factor (VIF) among the variables considered in the econometric models. As can be observed, the correlations between the explanatory variables are all less than 0.5, except the correlation between worker and age (−0.57). In addition, the VIF have values lower than 2.1. Therefore, multicollinearity problems do not arise. However, the correlations between risky assets holding ratio and the two another dependent variables are higher than 0.8, which indicates that they demonstrate similar behaviour in the sample under study. Therefore, in the following analysis, we focus only on risky assets ratio.

### 5.2. Econometric Analysis

The estimation of the models specified to contrast the hypotheses proposed are presented in Table 4, with risky assets ratio as the dependent variable. In Model 1, private health insurance and subjective risk tolerance have been included, which are positive and significant at 1% (*p*-value <0.01). This indicates that taking out PHI increases the portfolio risk of households compared to those who are uninsured, and that individuals with higher risk tolerance present greater portfolio risk with respect to those who claim to be risk-averse.

In order to contrast the second hypothesis regarding the moderating effect of subjective risk tolerance, two analyses have been carried out. On the one hand, Model 1 has been estimated for the subsamples of individuals with risk tolerance (SRT = 1) and for individuals with risk aversion (SRT = 0).

As we can see in both Model 2 and Model 3, the PHI variable is significant and positive, indicating that taking out health insurance increases families’ level of risk taking. However, the coefficient of Model 2 is lower than that of Model 3 (0.11 versus 0.26), which indicates that the effect is greater in the subsample of risk-averse households. In other words, in cases in which the individual is willing to take risks in their financial investments, taking out health insurance does not affect their risk-taking as much as it does risk-averse individuals, for whom the coverage provided by health insurance is more important. On the other hand, in Model 4—estimates for the total sample—we introduce the interaction between PHI and SRT. The results confirm that having PHI for risk-tolerant individuals reduces the percentage of risky assets in the portfolio relative to risk-averse individuals. That is, subjective risk tolerance exerts a moderating effect on the relationship between PHI and risk taking.

Regarding the control variables, in those variables where there is some consensus, the results are in line with previous evidence. Thus, the level of education, income and homeownership have a positive relationship with risk taking. On the contrary, gender reduces financial risk taking, with women being more risk averse. Employment status, specifically being active in the labour market, reduces risk taking. In this sense, it is noteworthy that a third of the sample is made up of retirees. Other variables are not significant in any of the models (marital status) or in some of them (health status in the subsample of SRT = 0 or children in SRT = 1).

### 5.3. Robustness Analysis

With the aim to analyse the robustness of the results, we carried out several analyses, considering different subsamples, alternative dependent variables and econometric methods. We re-estimated the models 2 (SRT = 1) and 3 (SRT = 0), but only reported here some of these, the rest being available upon request.

First, following Bucciol et al. [29], we have only re-estimated the models for the subsample of risky assets holders. The results regarding the PHI variable are positive and significant for two models (see Models 5 and 6 in Table A3 Appendix), with a higher coefficient when the sample is SRT = 1. Second, we considered a new dependent variable based on the risk level of each asset type. Specifically, similar to Li et al. [6], we created the portfolio risk average variable, computing the average of the share of bank accounts, bonds, mutual funds, and stocks and derivatives, weighting each asset by an increasing level of risk from 1 to 4 (see composition of this variable in Table A1 appendix). In Models 7 and 8 (Table A3 Appendix), we can see that the coefficients of PHI are positive and significant in both models, but that there is a big difference between these samples. Specifically, a coefficient of 0.2994 in the sample of SRT = 1 and a coefficient of 0.7636 for SRT = 0. Third, it is possible that marital status affects the result. According to Jianokoplos [26], for single households, there is a direct correspondence between the stated and observed risk measures. However, for married households, there is the possibility that the household member responding to the survey question regarding risk tolerance is not the household member responsible for making financial investment decisions for the household. Thus, we have re-estimated the models for the subsamples of married/partner and the singles households. The results (unreported by space) are similar in both subsamples. Fourth, with the purpose of comparing our results with previous studies, the models have been estimated for individuals aged 50 years or older. The results (unreported) are similar to the observations initially obtained. Fifth, we used the Heckman two-step model as an alternative method to control the zeros. The results of the second stage (non-reported by space) are similar to Tobit, but less significant. Sixth and last, we have estimated a simultaneous equations (3SLS) method to jointly estimate the equations for portfolio risk and health insurance. The results of estimations indicate that PHI is positive in both samples, but only significant when SRT = 0 (see Models 9 and 10 in Table A3 Appendix).

## 6. Discussion

The results obtained offer support for the arguments presented in hypothesis 1, according to which having PHI reduces the background risk derived from the uncertainty of medical expenses, which in turn contributes to taking financial risks. These results are robust to different dependent variables, subsamples and estimation methods. In addition, they are consistent with the limited empirical evidence obtained in other contexts, with very different healthcare systems, as well as cultural values. In this vein, to our knowledge, the only studies that have analysed the relationship between private health insurance and portfolio risk are Goldman and Maestas [2], Christellis et al. [7] and Li et al. [6]. Goldman and Maestas [2], in the context of the USA, found that individuals who face lower medical expenditure risk, as measured by their enrolment in a Medicare HMO or an employer supplemental insurance policy, are more likely to hold risky financial assets. Christellis et al. [7] found that Medicare eligibility has a significant impact on stockholding for college-educated households, but does not alter the financial risk-taking behaviour of households whose members have not finished college. Li et al. [6], using a sample obtained from a Chinese household survey, found that the use of health insurance with better policies is related to a higher probability of accepting risky assets. In the European context, the empirical evidence about household risk taking is scarce, and no study has analysed the effect of PHI in the risk taking. Christellis et al. [37] used the Survey on Health, Ageing and Retirement in Europe (SHARE) and focused on how the childhood conditions—such as socioeconomic status, cognitive abilities and health problems—influence portfolio choice. In Spain, Pinilla and López-Valcarcel [9], based on the Household Financial Survey (EFF), analysed the determinant of the private health insurance, but not household risk taking.

Regarding hypothesis 2, our results confirm the moderating effect of the subjective risk tolerance on the relationship between PHI and household risk taking. However, that effect is more significant in risk-averse individuals. Thus, for individuals with high tolerance to financial risk, having PHI positively influences the acquisition of risky assets, but to a lesser extent than in risk-averse individuals. The results are in accordance with Li et al. [6], the only work that has analysed this topic, although it is in reference to China, a country with a very different healthcare system and with cultural values that are far removed from those of Spain.

It is important to highlight that the EFF has the advantage of having enough observations for a study on the financial behavior of families, which is relevant given that only a small part of the population invests in financial assets, mainly high-income households [45]. For this reason, we believe that the results should be considered applicable to this type of family, which could be the one with the highest probability of contracting a private health insurance. However, Pinilla and González López-Valcárcel [9] find that, in Spain, there is no linear relationship between income and the probability of having private health insurance.

The work is not without limitations, all of them related to the source of information. The main one is the difficulty of accurately distinguishing the amount invested in the various financial assets since, in some cases, the EFF provides this data in aggregate form. For example, it is not possible to know the type of investment fund. This has made it difficult to create the variable portfolio risk average. In addition, it is impossible to know the type of health insurance contracted, as well as if the respondent is a civil servant or not. Finally, there exists the difficulty of carrying out a longitudinal study, since the EFF only contains a small number of common respondents in the different waves.

## 7. Conclusions

Our results show that PHI significantly increases household portfolio risk, especially in households with greater risk aversion, which represent 75% of the sample. Our paper contributes to the literature on household finance, providing a possible explanation for the fact that many households do not have investments in risky financial assets. In this sense, it stands out that the distribution of family wealth between real and financial assets shows an association to self-perceived health [46]. The results suggest that, in addition to improving public healthcare, families with resources should take out health insurance, allowing them to obtain better results from their financial investments by introducing riskier assets, which provide better returns.

Regarding the implications for public health, in our opinion, contracting private health insurance contributes directly to improving the quality of life of families whose income allows it. Likewise, it indirectly contributes to improving healthcare for other families with fewer resources, to the extent that by using private healthcare, they free up resources that can be used for other patients.

## Figures and Tables

**Table 1 ijerph-19-16248-t001:** A description of the control variables.

Variable	Values	Expected Relationship	References
Health status	From 1 to 5 1: bad/very bad 5: very good	Positive	[1,18,24,29,37]
Gender	1: women 0: men	Negative	[26,29,38,39]
Age	Log (n° years respondent in date of survey	Positive or negative	[2,29,39]
Educational level	Prim/Sec. Studies (1–0) Baccalaureate (1–0) Higher Studies (1–0)	Positive	[29,37,40,41,42]
Marital status	Married/Couple = 1 if respondent is married of cohabits with a partner and 0 otherwise	Positive, negative or non-significant	[1,17,19,29,37]
Presence of children	Dummy = 1 if the family has some children and 0 if no children	Negative	[18,22,29,36]
Labour status	Worker = 1 if respondent is employed or self-employed; = 0 otherwise	Positive or negative	[2,29,37]
Income level	Logarithm of household income in year previous of the survey	Positive	[34,43]
Home-ownership	Dummy = 1 if the house is owned of the family, and 0 otherwise	Positive or negative	[2,19,29,32]

1–0 Dummy variables. All variables refer to the respondent, except income and children, which correspond to the household.

**Table 2 ijerph-19-16248-t002:** Summary statistics.

	Mean	S.D.	Min	Max
*Dependent variables*				
Risky assets holding	0.3481	0.4764	0	1
Risky assets ratio	0.2054	0.3475	0	1
Portfolio share in stocks	0.1472	0.2902	0	1
*Explanatory and control variables*				
Private health insurance	0.3347	0.4719	0	1
Health status	2.8090	0.8200	1	4
Subjective risk tolerance	0.2451	0.4302	0	1
Women	0.3775	0.4848	0	1
Age (log years)	4.0642	0.2674	2.94	4.44
Baccalaureate	0.2780	0.4480	0	1
Higher studies	0.3605	0.4802	0	1
Married/Couple	0.6423	0.4793	0	1
Presence of children	0.4219	0.4939	0	1
Worker	0.4522	0.4977	0	1
Incomes (log)	10.554	0.8977	8.55	16.23
Home-owner	0.8386	0.3679	0	1

Frequency % in dummies and average in continuous variables.

**Table 3 ijerph-19-16248-t003:** The average values of risk taking with private health insurance and Subjective risk taking (SRT).

	Risky Assets Holding			Risky Assets Ratio			Portfolio Share in Stocks		
Sample	With PHI	Without-PHI	Chi2	With PHI	Without PHI	T-Test	With PHI	Without PHI	T-Test
All	0.5868	0.2280	771.62 ***	0.3706	0.1222	−28.00 ***	0.2639	0.0885	−23.24 ***
SRT = 1	0.7652	0.5117	104.68 ***	0.5596	0.3438	−10.33 ***	0.3795	0.2279	−8.20 ***
SRT = 0	0.4788	0.1662	477.07 ***	0.2563	0.0740	−20.94 ***	0.1939	0.0582	−17.77 ***

PHI: Private health insurance. SRT: Subjective Risk Tolerance, *** Significant at 1%.

**Table 4 ijerph-19-16248-t004:** Private health insurance and risk taking in Spain.

	Model 1		Model 2		Model 3		Model 4	
Sample	All Sample		SRT = 1		SRT = 0		All Sample	
	β	S.E.	β	S.E.	β	S.E.	β	S.E.
PHI	0.2021 ***	0.0241	0.1107 ***	0.0291	0.2623 ***	0.0286	0.2659 ***	0.0247
SRT	0.4353 ***	0.0208	-	-	-	-	0.5268 ***	0.0289
PHIxSRT	-	-	-	-	-	-	−0.1798 ***	0.0391
Health status	0.0321 ***	0.0125	0.0491 ***	0.0193	0.0229	0.0169 **	0.0317	0.0125
Women	−0.1037 ***	0.0216	−0.0697 **	0.0346	−0.1182	0.0288 ***	−0.1020	0.0216
Age (log years)	0.7267 ***	0.0548	0.7814 ***	0.0791	0.7250 ***	0.0772 ***	0.7240	0.0549
Baccalaureate	0.2386 ***	0.0270	0.1803 ***	0.0466	0.2557 ***	0.0354 ***	0.2312	0.0270
Higher studies	0.3811 ***	0.0270	0.3285 ***	0.0432	0.3985 ***	0.0365 ***	0.3749	0.0270
Married/Couple	−0.0182	0.0229	−0.0210	0.0350	−0.0244	0.0313	−0.0189	0.0230
With children	−0.0628 ***	0.0290	0.0079	0.0331	−0.1162 ***	0.0322 ***	−0.0633	0.0229
Worker	−0.1176 ***	0.0243	−0.1373 ***	0.0345	−0.1178 ***	0.0344 ***	−0.1204	0.0243
Incomes (log)	0.1970 ***	0.0133	0.1394 ***	0.0177	0.2533 ***	0.0199 ***	0.1979	0.0133
Home owner	0.1488 ***	0.0305	0.0962 **	0.0429	0.1875 ***	0.0439 ***	0.1506	0.0306
Constant	−5.7581 ***	0.2536	−4.8111 ***	0.3440	−6.4199 ***	0.3726 ***	−5.7760	0.2536
Observations	6110		1498		4612		6110	
Left-censored obs.	3983		536		3447		3983	
Pseudo R^2^ (%)	30.33		24.47		23.04		30.55	
Log-likelihood	−3298.44		−1033.52		−2267.60		−3287.86	

Dependent variable: Risky assets ratio. Estimation method: Tobit. ***, ** Significant at 1%, 5%, respectively.

## Data Availability

Publicly available datasets were analysed in this study. The data can be found here: https://pas.bde.es/privbde/es/pas/eff-datos/index2017.html (accessed on 15 January 2021).

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
