# Peer review of "Private Health Insurance and Financial Risk Taking in Spain—The Moderating Effect of Subjective Risk Tolerance"

_ijerph, 2022, doi:10.3390/ijerph192316248_

Round 1

Reviewer 1 Report

The article is well written and provides an interesting analysis.

The introduction talks about “coverage of medical expenses” and PHI as “reducing the risk derived from health expenses”. However, the authors do not discuss in section 2 specificities of the Spanish system that go in the opposite direction: Prescriptions from the NHS are covered 60% or 100% depending on age, whereas the insured bears the complete costs of medicine prescription from PHI. In this respect, while PHI reduces the risk of some health expenditures, it increases the risk of expenditure in medicines. There is also no talk regarding the insurance system of legal servants (MUFACE) which shares elements of both public and private systems and how is it considered. If it is possible to control for being a civil servant in the analysis it would help. If it is not possible, it should be acknowledged as a limitation and born in mind in the interpretation of results.

Question P 5.22 is quite vague since it only talk about whether any type of health insurance is contracted. It could be, for instance, a dental policy. While to a certain extent the argument of the paper can be made with any type of policy, the authors should acknowledge as a limitation of the article this lack of specificity.

Regarding interpretation of results, the authors as making a “causal” interpretation of the results that might not be warranted based on cross-sectional data. They should be explicit on this in the section on limitations.

Author Response

Dear Reviewer,
Thank you very much for taking the time to review our article and for your suggestions. Please find our reply below:

Reviewer suggestion:

The introduction talks about “coverage of medical expenses” and PHI as “reducing the risk derived from health expenses”. However, the authors do not discuss in section 2 specificities of the Spanish system that go in the opposite direction: Prescriptions from the NHS are covered 60% or 100% depending on age, whereas the insured bears the complete costs of medicine prescription from PHI. In this respect, while PHI reduces the risk of some health expenditures, it increases the risk of expenditure in medicines. There is also no talk regarding the insurance system of legal servants (MUFACE) which shares elements of both public and private systems and how is it considered. If it is possible to control for being a civil servant in the analysis it would help. If it is not possible, it should be acknowledged as a limitation and born in mind in the interpretation of results.

Authors ‘response:

According to your suggestion regarding the increased risk of drug spending, we consider that there are several reasons that make it no so relevant: (1) The fact that private health insurance is optional makes that the trade-off between advantages and disadvantages is already implicit in the decision to take out a policy; (2) In Spain, it is possible (and usual) to ask the family doctor to prescribe the patient a medicine that has been prescribed by a private specialist, so that the cost is covered by the NHS; (3) The main companies offering health insurance (e.g., Sanitas, Adeslas, Mapfre), offer policies with coverage of pharmacy expenses or reimbursement of medication expenses, with an annual limit. This is a complementary coverage and therefore influences the price of the insurance. However, the information available does not allow us to know this data, so it has been considered a limitation of the study. An analysis of the characteristics of this coverage can be consulted in:

https://www.mvaseguradores.com/noticias/otros-seguros/gastos-farmacia-seguro-salud/#sp_6

Regarding your suggestion about the case of civil servants through MUFACE, we have included in section 2 a reference to this case. However, the EFF does not allow distinguishing whether the employee with a permanent contract is a civil servant or not. Following your suggestion, we have included this question in the limitations of the study.

Reviewer suggestion:

Question P 5.22 is quite vague since it only talk about whether any type of health insurance is contracted. It could be, for instance, a dental policy. While to a certain extent the argument of the paper can be made with any type of policy, the authors should acknowledge as a limitation of the article this lack of specificity.

Authors ‘response:

According the information provided by the EFF does not specify the type of health insurance taken out, so, following their suggestion, this has been considered as a limitation of the study.

Reviewer suggestion:

Regarding interpretation of results, the authors as making a “causal” interpretation of the results that might not be warranted based on cross-sectional data. They should be explicit on this in the section on limitations.

Authors ‘response:

Following your suggestion, a reference to the use of data referring to a single year has been included among the limitations. The EFF is carried out every three years, but the selection of the sample and of the responses obtained makes it difficult to produce a panel of data, as there is a significant loss of observations. In any case, it would be possible to repeat the study on the basis of previous editions, which can be considered as an extension of the study.

Reviewer 2 Report

I enjoyed reading this paper on financial risk-taking and the Spanish health care system.  In my view, the authors have contributed to the literature in this area, and indeed, have engaged well with this literature.  Their approach is diligent and appropriate, and their work provides some interesting insights.

For the sake of brevity, I’ll cut-to-the chase re: a few points that the authors may wish to consider:

·        The sample size – 6,000 plus – is impressive.  Yet, what is not obvious is whether the income and wealth distribution(s) of this sample mirrors that of the Spanish population.  I suspect that those holding financial assets of the type described by the authors will be generally wealthier households.  Indeed, one is left wondering whether the queue-jumping the authors describe in their paper is the preserve of the wealthy.

·        If the income/wealth distribution(s) in the sample differ from those of the wider population, I feel that the authors should acknowledge this, and reflect on the implications for their results.

·        Perhaps, if they have space, the authors could further develop their thinking on the public health implications of their results.

·        The authors do not cite any previous related work in this area previously published in the International Journal of Environmental Research and Public Health.  I feel that it’s appropriate for the authors to consider citing such relevant publications, as it demonstrates an engagement with the journal, the literature, and its readership.

Overall, this is a fine submission, and I recommend publication subject to those minor points identified above.

Author Response

Dear Reviewer,
Thank you very much for taking the time to review our article and for your suggestions. Please find our reply below:

Reviewer suggestion:

The sample size – 6,000 plus – is impressive.  Yet, what is not obvious is whether the income and wealth distribution(s) of this sample mirrors that of the Spanish population.  I suspect that those holding financial assets of the type described by the authors will be generally wealthier households.  Indeed, one is left wondering whether the queue-jumping the authors describe in their paper is the preserve of the wealthy.

Authors´ response:

Indeed, as we have added in the work, the sample is characterized by an over-representation of wealthy households, which is important for a study referring to financial risk taking. In addition, in the sample selection we have excluded households that do not declare having financial investments.

Reviewer suggestion:

If the income/wealth distribution(s) in the sample differ from those of the wider population, I feel that the authors should acknowledge this, and reflect on the implications for their results.

Authors´ response:

The objective of the study focuses on financial risk taking of families, so the selection of the sample itself indicates that the results are applicable to households with a certain level of income. However, following your suggestion, we have included a comment about it in the discussion of results.

Reviewer suggestion:

Perhaps, if they have space, the authors could further develop their thinking on the public health implications of their results.

Authors´ response:

Following your suggestion, we have included a new paragraph in the conclusions about the public health implications of our results.

Reviewer suggestion:

The authors do not cite any previous related work in this area previously published in the International Journal of Environmental Research and Public Health.  I feel that it’s appropriate for the authors to consider citing such relevant publications, as it demonstrates an engagement with the journal, the literature, and its readership.

Authors´ response:

We have carried out several searches in google scholar, papers published on International Journal of Environmental Research and Public Health, by different keywords (health insurance, private health insurance, financial risk, financial) in the tittle. Following your suggestion, we have cited in the conclusions the paper of López-Casasnovas and Saez (2020), published on this journal.

References:

López-Casasnovas, G., & Saez, M. (2020). Saved by wealth? Income, wealth, and self-perceived health in Spain during the financial crisis. International Journal of Environmental Research and Public Health17(19), 7018.

Reviewer 3 Report

It's a very interesting paper and I enjoyed my reading. However, I'd like to propose my concern regarding the research design. The purpose of this study is to measure the effect of private health insurance on financial risk taking. But we know that enrolling in private health insurance is voluntary, which may lead to endogeneity in causal inference. I wonder how did the authors deal with this problem? Besides, as the authors mentioned, "Spain is a good laboratory since it has public health coverage that could eliminate the effect of PHI on risk taking", how did you measure the real or pure effect of PHI under such a public health coverage system?

Author Response

Dear Reviewer,
Thank you very much for taking the time to review our article and for your suggestions. Please find our reply below:

Reviewer suggestion:

It's a very interesting paper and I enjoyed my reading. However, I'd like to propose my concern regarding the research design. The purpose of this study is to measure the effect of private health insurance on financial risk taking. But we know that enrolling in private health insurance is voluntary, which may lead to endogeneity in causal inference. I wonder how did the authors deal with this problem? Besides, as the authors mentioned, "Spain is a good laboratory since it has public health coverage that could eliminate the effect of PHI on risk taking", how did you measure the real or pure effect of PHI under such a public health coverage system?

Authors ‘response:

Regarding your comment about the control of endogeneity derived from reverse causality, we have re-estimated the model with simultaneous equations (3SLS). The results are in models 9 and 10 in table A3 of Appendix. As we have indicated in the study, the model shows that PHI maintains the positive sign in both models, although it is only significant in the SRT=0 subsample, which corroborates the moderating effect of subjective risk tolerance on the relationship between PHI and financial risk taken.

In response to your question about measuring the real or pure effect of PHI, we have to acknowledge that it has not been possible to make such a measurement, as we do not have the information to do so. We have assumed that the Spanish context is very different from countries such as the United States where most of the studies on the subject have been carried out.

Reviewer 4 Report

A great manuscript! I have made a few suggestions by highlighting the relevant text and inserting notes on the LHS. (Ignore the highlighted title Discussion).

One thing that puzzles me a bit is that you have a detailed section on the literature review which is well written, and yet I would have thought that you would include the lit rev as part of the Method, with themed sections reported on in the Results. However, if the journal accepts this style of first reporting on the lit rev, and then developing the Method & Results, it's also acceptable. It certainly lightens the Results section and enables the reader to focus only on the model outcomes.

I also note that you have written the Methodology in the present, rather than the past tense: "...the study focuses..." not "the study focused...". Both are acceptable, although to be totally precise, the study is completed and the past tense would be better. Yet in the Variables section, you wrote "To contrast the 2nd hypothesis, we classified..." i.e. past tense. I would standardize the tenses, even though it doesn't make a material difference to the outcomes, but improves the style of writing.

Author Response

Dear Reviewer,
Thank you very much for taking the time to review our article and for your suggestions. Please find our reply below:

Reviewer suggestions:

A great manuscript! I have made a few suggestions by highlighting the relevant text and inserting notes on the LHS. (Ignore the highlighted title Discussion).

One thing that puzzles me a bit is that you have a detailed section on the literature review which is well written, and yet I would have thought that you would include the lit rev as part of the Method, with themed sections reported on in the Results. However, if the journal accepts this style of first reporting on the lit rev, and then developing the Method & Results, it's also acceptable. It certainly lightens the Results section and enables the reader to focus only on the model outcomes.

I also note that you have written the Methodology in the present, rather than the past tense: "...the study focuses..." not "the study focused...". Both are acceptable, although to be totally precise, the study is completed and the past tense would be better. Yet in the Variables section, you wrote "To contrast the 2nd hypothesis, we classified..." i.e. past tense. I would standardize the tenses, even though it doesn't make a material difference to the outcomes, but improves the style of writing.

Authors ´response:

We have considered most of the suggestions included in the pdf document (attached with our responses).
As for the structure of the article, we prefer the literature review section to appear separately in order to make the hypotheses clear.
We have homogenised the verb tenses, as you requested.

Round 2

Reviewer 3 Report

Please state your limitations in terms of my question about measuring the pure effect of PHI in Spanish setting.

Author Response

Dear Reviewer,
Thank you very much for taking the time to review our article and for your suggestions. Please find our reply below:

Reviewer suggestion:

Please state your limitations in terms of my question about measuring the pure effect of PHI in Spanish setting

Authors´ response:

In response to your question, we consider that, in Spain, the existence of a NHS with universal coverage makes that having a private health insurance implies an additional expense that is not strictly necessary. For this reason, according to Pinilla and López-Valcarcel (2020) (cite included in the new version), in Spain, the private health insurance is supplementary or alternative to the coverage offered by public health. In other countries like the USA, contracting private health insurance becomes a necessity for a large part of the population.